# REGRESSOR-FREE INTERMEDIATE LAYER DISTILLATION VIA TEACHER PRUNING

## ABSTRACT

In deep learning, knowledge distillation from a teacher model to a smaller student model is commonly utilized to develop compact models without significantly sacrificing performance. Among various approaches, intermediate layer distillation often encounters a dimensional discrepancy between the teacher's layer and the corresponding student's layer. Typically, this problem is addressed by introducing a regressor to the student's target layer to forcibly align dimensions. However, because distillation through the regressor transfers knowledge indirectly, the teacher's knowledge is not fully transmitted to the student. Through preliminary experiments using the probing method, we first investigate whether the regressor effectively transfers the intended knowledge and reveal suboptimality in the conventional regressor-based method. Motivated by these findings, we propose an alternative method inspired by pruning techniques that directly adjusts the teacher's target layer dimensions to match those of the student. Extensive experiments on diverse backbone architectures and datasets demonstrate that our method consistently achieves superior accuracy compared to conventional regressor-based approaches. In particular, for ResNet models trained on CIFAR-100 and TinyImageNet, the student trained with our method attains accuracies of 77.50% and 49.23%, respectively, even exceeding the teacher's accuracy.

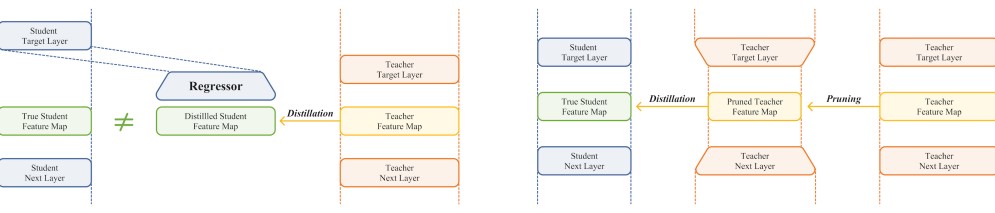

(a) Traditional ILD method.    (b) Our method using teacher pruning.

Figure 1: Our proposed method, compared to traditional ones. With our method, we can directly distill the intermediate knowledge of teacher to student.

## 1 INTRODUCTION

Deep neural networks have advanced rapidly and now demonstrate high performance in diverse domains and are increasingly being deployed in real-world applications. However, modern deep neural networks have grown so large that they incur prohibitively high costs in memory, computation time, and power consumption. This issue hinders the use of DNNs in devices and raises concerns about substantial power consumption that can contribute to environmental pollution (Strubell et al., 2020).

To address these issues, a variety of model compression techniques have been studied. These include pruning, which removes less influential components of a model; knowledge distillation, which transfers knowledge from a larger teacher model to a smaller student model so that the student can also achieve competitive performance; and quantization, which reduces storage space by representing model parameters with reduced precision (Choukroun et al., 2019).

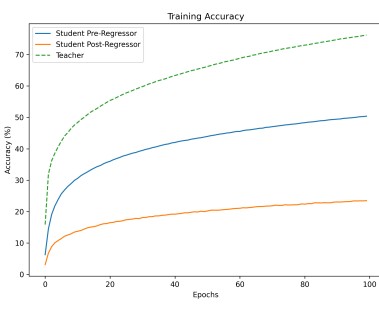 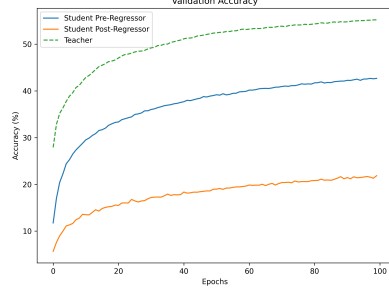

(a) Probing results on a training set      (b) Probing results on a validation set

Figure 2: Probing results on feature map of a teacher and pre/post-regressor of a student.

Among these methods, knowledge distillation (KD) extracts useful knowledge from a large, high-performing teacher network and uses it to train a student network, which is often designed to be smaller and more efficient, thereby greatly improving its performance. KD is commonly categorized into logit distillation (LD) and intermediate layer distillation (ILD). LD is the most fundamental KD method in which the student model learns to mimic the teacher's final output distribution. ILD, on the other hand, focuses on extracting and distilling information in the intermediate layers, based on the idea that useful knowledge resides not only in the final logits but also within intermediate representations. In this process, unlike LD, the intermediate layer dimensions of the teacher and student networks often differ. Since the distillation ultimately aims to make the two target layers exhibit similar features, it is essential to align their dimensions. This alignment is commonly achieved using the method proposed in Romero et al. (2014), which introduced ILD. In that work, a "regressor" layer, which is typically a single fully connected layer, is attached to the target student layer to forcibly align the sizes of the two layers. Many ILD variants handle the width mismatch by introducing a student-side regressor as in Romero et al. (2014), while the implications of this choice have received limited analytical attention. Regressor-based ILD, however, has a fundamental limitation: **the student receives the teacher's knowledge only indirectly via the regressor**, rather than directly from the teacher.

Through a preliminary experiment using the probing method (Alain & Bengio, 2016), we observed that ILD employing a regressor can be suboptimal. The probing technique is a method for analyzing feature maps by attaching a shallow classification layer to a frozen feature map and training it to determine whether a specific attribute is present in that feature map. Figure 2 shows the results of our preliminary experiment. After training the student using the method proposed in FitNet (Romero et al., 2014), we applied the probing method to the feature map of the target layer of the teacher and to the feature maps of the student layers immediately before and after the regressor, respectively, to investigate the amount of information each contained. The results showed that probing on the student's post-regressor feature map, which receives distillation directly from the teacher, achieved poorer performance than the others. This suggests that the post-regressor feature map, which directly receives the teacher's information through distillation, does not adequately absorb the teacher's information. These observations provide the motivation to investigate the suboptimality of the regressor-based ILD approach and to regard the regressor itself as a primary cause. Further analysis of the probing results is presented in Section 4.3.1.

In this paper, we introduce a **Regressor-free Intermediate Layer Distillation** method that utilizes pruning to match the dimensionality between the teacher and the student. Rather than inserting an extra regressor layer to resolve the dimensional mismatch between the intermediate layers of the teacher and the student, we directly prune the target layer of the teacher so that its dimensionality matches the student's.

Figure 1 compares the conventional ILD method with our method. The conventional approach uses a regressor to resolve the dimension mismatch, resulting in indirect distillation of the teacher's feature map. Furthermore, the regressor is employed only for distillation and is not used at inference time, which means that the teacher's feature map does not correspond to the student's true feature map. In contrast, our method resolves the feature map dimension mismatch by pruning the teacher, directly transferring the teacher's knowledge and making it fully available for the student to use.

The main contributions of this work are threefold:

1. We propose a novel Regressor-free Intermediate Layer Distillation method, an intermediate layer distillation framework that uses pruning to resolve the teacher-student dimensional mismatch problem.

2. Through extensive experiments on a variety of image classification models and datasets, we demonstrate that our method consistently outperforms conventional intermediate layer distillation approaches.

3. We conduct a comprehensive analysis from both theoretical and experimental perspectives to explain why our method delivers superior performance.

## 2 RELATED WORKS

### 2.1 KNOWLEDGE DISTILLATION

Knowledge distillation is a model compression technique that extracts meaningful knowledge from a large teacher network and uses it to train a smaller student network, allowing the student to approach the performance of the teacher. The idea was first introduced by Hinton et al. (2015). Broadly, knowledge distillation can be divided into two categories: logit distillation and intermediate layer distillation.

Logit distillation transfers information from the teacher's final outputs in various ways. Following Hinton et al. (2015), many LD method have been proposed for various purposes and using diverse techniques (Zhang et al., 2018; Mirzadeh et al., 2020; Zhao et al., 2022; Sun et al., 2024).

Intermediate layer distillation transfers information from intermediate feature representations. Romero et al. (2014) were the first to propose feature map based ILD. The teacher's target feature map is provided as a hint for the corresponding student layer to mimic. Subsequently, numerous studies have exploited various forms of information from the intermediate layer (Tung & Mori, 2019; Heo et al., 2019; Tian et al., 2019).

In conventional ILD, the dimensions of the teacher and student target feature map often differ. Generally, this problem is addressed either by simply adding a small fully connected layer to the student's target layer to forcibly align the dimensions, or by indirect methods based on data statistics. In this paper, we solve the dimension mismatch issue using pruning, proposing a method that directly distills the teacher's information without employing a regressor.

### 2.2 PRUNING

Pruning is a model compression technique that reduces model size by removing parts deemed unimportant while preserving performance as much as possible. Modern pruning was first introduced by Han et al. (2015) and has since been further developed. Pruning is broadly divided into unstructured pruning and structured pruning.

Unstructured pruning removes parameters regardless of the architectural structure, simply setting the selected weights to zero. Although this approach is easy to implement and train, the zeroed weights continue to occupy memory and incur computations when specialized hardware is not used, so its practical speed and memory saving benefits are limited (Han et al., 2015; Frankle & Carbin, 2018; Wang et al., 2020; Tanaka et al., 2020).

Structured pruning is a method that prunes at the level of structural units, and by actually removing parts of the network, it yields practical improvements in memory usage and speed. However, compared to unstructured pruning, the pruning process is more complex and the performance drop is larger relative to the pruning ratio (Crowley et al., 2018; Lee et al., 2024; Liu et al., 2025). After Li et al. (2016) first introduced structured pruning in convolutional networks, various works have proposed structured pruning method using diverse criteria (Luo et al., 2017; Liu et al., 2017; Hu et al., 2016).

Pruning is sometimes employed not only as a model compression tool but also as a means to achieve other objectives. In such cases, researchers typically choose unstructured pruning because neither

the size of the model nor the inference time is their goals, making the usual drawbacks of unstructured pruning negligible (Han et al., 2016; Mallya & Lazebnik, 2018; Hintersdorf et al., 2024).

This paper likewise employs pruning for a purpose other than model compression. The teacher network is pruned, but only a small portion of a specific layer is removed, leaving the overall model size nearly unchanged and its performance intact, while producing a teacher that can directly distill knowledge to the student.

### 2.3 Knowledge Distillation with Pruning

There are also papers that use pruning and distillation simultaneously. In most of these studies, the model before pruning serves as a teacher and the model after pruning serves as a student, and distillation is performed to preserve performance after pruning (Wang et al., 2023; Pan et al., 2024; Schmitt et al., 2024). On the other hand, Park & No (2022) reported that the student distilled from the pruned teacher shows better performance than the student distilled from the original teacher. This result is consistent with the experimental findings of the present paper.

This paper is not the first work to prune the teacher network to align its dimensionality with that of the student. Sarridis et al. (2025) attempted an intermediate layer distillation approach based on curriculum learning that preserves the information flow path and noted that teacher pruning could be performed in cases where the teacher-student dimensions do not match. However, in that work, pruning was introduced only as an auxiliary step within the overall pipeline, rather than as a primary methodology for ILD, and the design or independent effectiveness of teacher pruning for ILD was not analyzed in depth. In contrast, this paper experimentally demonstrates that the presence of a regressor in ILD process hinders learning, and proposes teacher pruning as a means to resolve this issue. Through extensive experiments, we further show that this approach is effective. In this respect, the goals and contributions of the two papers are entirely different. To the best of our knowledge, no previous work has identified the regressor problem in ILD and proposed pruning as a solution.

## 3 Methodology

In this study, we propose a novel Regressor-free Intermediate Layer Distillation method, which prunes the teacher network to resolve the dimension mismatch problem in ILD. Our method works in two main steps: (1) pruning the teacher's target layer, and (2) training the student model by distilling features from the pruned intermediate layer of the teacher.

Earlier studies on ILD (Romero et al., 2014; Zhang et al., 2017; Jiao et al., 2019) handle the size mismatch between the feature map of the teacher and the student by adding an additional regressor layer (often a 1×1 convolutional layer) to the student's side. However, as discussed in Section 1, the regressor layer may constitute an unnecessary step.

To address this inefficiency issue, instead of using the regressor, we prune the teacher's target layer so that its size matches the student's directly. Our approach reduces unnecessary information loss during ILD and helps knowledge distillation to be carried out more efficiently.

### 3.1 Teacher Pruning

During the teacher pruning stage, we first select a target layer at which ILD will be performed. We select the target layer that lies at the same depth (*e.g.*, within the corresponding block) in both the teacher and the student networks. Let the feature map size of the teacher and student target layers be $f_t$ and $f_s$, respectively. Only the target layer of the teacher is pruned so that its feature map size is reduced from $f_t$ to $f_s$, while all other layers in the teacher remain untouched. Specifically, to minimize the number of layers and parameters affected, we modify only the output size of the layer immediately before the target feature map and the input size of the layer immediately after it. As a result, the total parameter count of the pruned model is nearly identical to that of the original model, and the performance gap between the pruned teacher and the original teacher remains negligible. This means that the information removed during pruning process has little effect on the model's core knowledge. For pruning, we adopt the simple yet effective $L_1$ norm based channel pruning technique: channels with the smallest $L_1$ norms in the target layer are removed until the desired dimension is reached. After pruning, a retraining process is conducted to restore the original

performance. Retraining is performed on the same dataset used to train the teacher, but for fewer epochs.

## 3.2 DISTILLATION TO STUDENT

Because the teacher's pruned target layer and the student's target layer are already dimensionally aligned, distillation can be performed directly without any additional processing. The distillation objective employs a cosine similarity loss that minimizes the angular difference between the teacher's and student's feature map. Specifically, the distillation loss is defined as:

$$\mathcal{L}_{\text{KD}} = 1 - \frac{1}{B} \sum_{b=1}^{B} \cos\left(f_S^{(b)}, f_T^{(b)}\right), \tag{1}$$

where $B$ denotes the batch size, $f_S^{(b)}$ and $f_T^{(b)}$ denotes the vectorized feature maps of the student and teacher, respectively, for the $b$-th sample.

The overall loss function combines the standard classification loss and the distillation loss:

$$\mathcal{L} = \alpha \mathcal{L}_{\text{cls}} + (1 - \alpha)\mathcal{L}_{\text{KD}}. \tag{2}$$

The coefficient $\alpha$ balances the two loss terms. Unless otherwise noted, we set $\alpha = 0.1$. An ablation study on the choice of $\alpha$ values is provided in Appendix A.3.1.

## 3.3 MUTUAL INFORMATION BASED RATIONALE

Using a mutual information analysis (Shannon, 1948; Shwartz-Ziv & Tishby, 2017), we can show that the performance of our method is lower bounded by that of the traditional, regressor-based ILD method.

Mutual information quantifies the amount of information shared between two random variables $X$ and $Y$, and is given by:

$$I(X;Y) = \sum_{x \in \mathcal{X}} \sum_{y \in \mathcal{Y}} p(x,y) \log \frac{p(x,y)}{p(x)p(y)}. \tag{3}$$

By treating both feature maps as random variables, we first compute the mutual information between the student's feature map and that of the pruned teacher. We then compare this value with the mutual information between the regressor-transformed student feature map and the original teacher feature map used in conventional ILD. We prove that the mutual information in our scheme is lower bounded by that of the conventional ILD method. Consequently, we aim to show that our approach is guaranteed, at least theoretically, to perform no worse than the traditional ILD baseline.

Let the teacher's feature map be $f_t$, the student's feature map $f_s$, and the pruned teacher's feature map $f_{tp}$. After pruning, some information is inevitably lost, which we can express as:

$$I(f_t; f_s) - I(f_{tp}; f_s) \leq \gamma, \tag{4}$$

where $\gamma$ denotes the information loss introduced by pruning. Our experiments show almost no performance gap between the original and pruned teachers, so $\gamma$ can be regarded as negligibly small.

Moreover, for the conventional ILD method, the mutual information after the regressor $R(\cdot)$ is applied satisfies, by the data processing inequality (Cover, 1999),

$$I(f_t; R(f_s)) \leq I(f_{tp}; f_s). \tag{5}$$

Combining Eq. 4 with Eq. 5 yields:

$$I(f_{tp}; f_s) \geq I(f_t; R(f_s)) - \gamma, \tag{6}$$

which indicates that the mutual information achieved by our method is at least as large as that of the conventional ILD approach, up to the small error term $\gamma$.

# 4 EXPERIMENTS

## 4.1 EXPERIMENTAL SETTINGS

### 4.1.1 DATASETS

We conducted experiments on the widely used vision benchmarks, CIFAR-100 (Krizhevsky et al., 2009) and TinyImageNet (Le & Yang, 2015). CIFAR-100 is an image dataset of 32 × 32 pixels, containing 50,000 training images and 10,000 test images, and is composed of 100 classes. Tiny-ImageNet is a subset of ImageNet (Deng et al., 2009) and consists of 100,000 training images and 10,000 validation images, across 200 classes. we evaluated on the validation split. To reduce training time, TinyImageNet images were downsized from their original 64×64 resolution to 32×32, which is the same resolution as CIFAR-100.

### 4.1.2 MODELS

For architectures, we used ResNet (He et al., 2016), VGG (Simonyan & Zisserman, 2014), and ShuffleNetV2 (Ma et al., 2018), which are widely used architectures for the image classification. For ResNet, we used ResNet101 as the teacher model and ResNet18 as the student model. The VGG variants inherently have identical feature map dimensions across models, so the teacher–student feature map dimensions are identical even without pruning. Therefore, we used the VGG16x4 model, whose width is increased four times, as the teacher model to force the dimension of feature map to differ from those of the student. As a result, we used VGG16x4 as the teacher model and VGG11 as the student model. In ShuffleNetV2, pruning only the desired layer is complicated due to its skip connections. To avoid this, we slightly modified the ShuffleNetV2 architecture by inserting convolutional layers in place of skip connections. The modified ShuffleNetV2 model showed negligible performance differences compared with the original. We used ShuffleNetV2x2.0 as the teacher model and ShuffleNetV2x0.5 as the student model. In all experiments, both teacher and student models were trained from scratch. Implementation details are described in Appendix A.1. *We share our source code for model implementation and experimentation as an anonymized repository during the review process*[1].

### 4.1.3 BASELINES

We conducted five types of student experiments: no distillation, logit distillation, traditional ILD using a regressor, logit distillation from the pruned teacher, and ILD from the pruned teacher. Here, no distillation, logit distillation, and ILD using a regressor serve as the main baselines for comparison with our method. The ILD using the regressor followed the FitNet method by Romero et al. (2014). Our main goal is to outperform the three baselines mentioned above and, in particular, to demonstrate that our method surpasses the conventional ILD approach, which employs a regressor, in most cases. Logit distillation from the pruned teacher follows the method of Park & No (2022), which achieves better distillation performance simply by using a pruned teacher instead of the original teacher. This baseline serves to confirm that the performance gain of our method arises not only from the employing a pruned teacher but also from eliminating the regressor in conventional ILD methods. Finally, ILD from the pruned teacher is our proposed method.

Note that we are not seeking to surpass the latest state-of-the-art performance. Rather, we aim to highlight issues with the regressor mechanism typically used in standard ILD. Consequently, we did not compare against the latest methods and instead focused our experiments on the presence or absence of the regressor and on whether the teacher was pruned or not. Nevertheless, our approach is versatile because it can be readily applied to other ILD frameworks if teacher pruning is available. That is, our method can be combined with other ILD techniques using a regressor to obtain performance improvements.

## 4.2 MAIN RESULTS

Table 1 shows our main experimental results. Several interesting observations can be made from these results. All experiments were conducted five times and the average value is reported.

---

[1]https://anonymous.4open.science/r/Teacher-Pruning-KD

Table 1: Experimental results on CIFAR-100 and TinyImageNet. We evaluated ResNet, VGG, and ShuffleNetV2. In the Architecture column, T and S indicate teacher and student networks. Target layer denotes the pruning layer in the teacher, equivalently the distillation source for the student. Baseline refers to the unpruned model. Lower layer numbers indicate shallower pruning, higher numbers deeper pruning. On the student side, four settings are reported: w/o KD (no distillation), LD (logit distillation from the original or pruned teacher at the target layer), FitNet (ILD from the unpruned teacher), and our method (distillation from the teacher pruned at the same target layer). The best accuracy in each case is shown in **bold**.

| Dataset | Architecture | Target Layer | Teacher | Student | | | |
|---|---|---|---|---|---|---|---|
| | | | | w/o KD | LD | FitNet | Ours |
| **CIFAR-100** | T-ResNet101 S-ResNet18 | Baseline | 76.97 | 74.79 | 75.46 | - | - |
| | | Layer1 | 76.94 | - | **75.69** | 74.76 | 74.64 |
| | | Layer2 | 76.73 | - | **75.71** | 75.02 | 73.39 |
| | | Layer3 | 76.45 | - | 75.89 | 75.32 | **76.49** |
| | | Layer4 | 77.54 | - | 75.34 | 75.40 | **77.50** |
| | T-VGG16x4 S-VGG11 | Baseline | 74.35 | 67.71 | 68.46 | - | - |
| | | Layer1 | 73.88 | - | **69.07** | 68.40 | 68.70 |
| | | Layer2 | 73.62 | - | 69.15 | 68.27 | **70.34** |
| | | Layer3 | 74.20 | - | 69.18 | 69.09 | **70.59** |
| | | Layer4 | 74.32 | - | 69.16 | 68.93 | **69.45** |
| | | Layer5 | 74.02 | - | 69.03 | 68.95 | **69.84** |
| | T-ShuffleNetV2x2.0 S-ShuffleNetV2x0.5 | Baseline | 65.39 | 57.34 | 57.84 | - | - |
| | | Layer1 | 63.15 | - | **57.79** | 57.45 | 57.63 |
| | | Layer2 | 65.22 | - | 58.37 | 57.95 | **59.08** |
| | | Layer3 | 65.10 | - | 57.87 | 58.02 | **59.04** |
| | | Layer4 | 65.38 | - | **59.06** | 58.34 | 58.81 |
| **TinyImageNet** | T-ResNet101 S-ResNet18 | Baseline | 46.14 | 44.49 | 45.47 | - | - |
| | | Layer1 | 48.02 | - | 45.37 | 44.93 | **45.54** |
| | | Layer2 | 47.93 | - | **45.68** | 44.98 | 44.63 |
| | | Layer3 | 47.70 | - | 45.27 | 45.29 | **47.04** |
| | | Layer4 | 47.19 | - | 45.37 | 45.08 | **49.23** |
| | T-VGG16x4 S-VGG11 | Baseline | 44.33 | 39.27 | 40.31 | - | - |
| | | Layer1 | 40.88 | - | **40.67** | 40.00 | 39.94 |
| | | Layer2 | 40.12 | - | 40.55 | 40.31 | **40.95** |
| | | Layer3 | 40.19 | - | 40.53 | 40.15 | **40.71** |
| | | Layer4 | 40.91 | - | 40.59 | 40.47 | **42.85** |
| | | Layer5 | 40.33 | - | 40.86 | 40.26 | **43.51** |
| | T-ShuffleNetV2x2.0 S-ShuffleNetV2x0.5 | Baseline | 35.03 | 30.25 | 28.16 | - | - |
| | | Layer1 | 33.56 | - | 28.43 | 27.89 | **28.64** |
| | | Layer2 | 34.74 | - | 27.89 | 27.79 | **28.86** |
| | | Layer3 | 34.45 | - | 28.10 | 27.63 | **30.45** |
| | | Layer4 | 36.01 | - | 29.87 | 27.62 | **31.62** |

Even after teacher pruning, performance remains nearly the same as that of the original teacher. This is because pruning is applied to only a very small portion of the network, so the total number of parameters remains similar to that of the original teacher. In practice, the pruned teacher performs comparably to the original teacher. In most cases that performance drop occurs, it remains within 1%p of the original teacher, except for VGG network on TinyImageNet. Furthermore, in many cases, performance even increases after pruning. This phenomenon has also been observed in many other pruning papers (Han et al., 2015; Frankle & Carbin, 2018; Luo et al., 2017). These results also provide concrete evidence that the value $\gamma$ in Eq. 6 is negligible in most cases, which in turn means that our method is at least as good as the traditional ILD approach.

When logit distillation is performed from the pruned teacher, performance is generally higher than when logit distillation is performed from the original teacher. This result is consistent with what was reported by Park & No (2022). However, because the pruning ratio is extremely small, the performance improvement over distillation from the original teacher is modest, and in some cases,

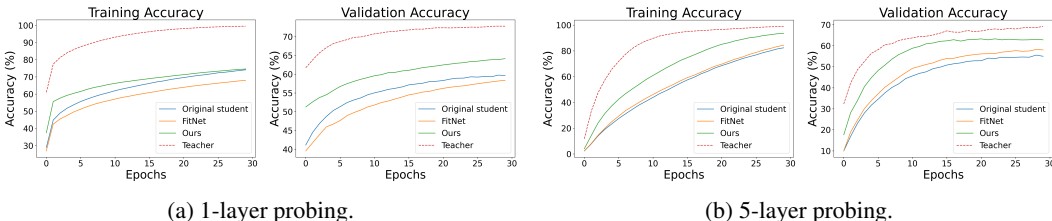

(a) 1-layer probing.            (b) 5-layer probing.

Figure 3: Probing results

the performance even decreases slightly. Park & No (2022) explained that this phenomenon occurs because pruning the teacher acts as a form of regularization. The pruned teacher's output distribution is smoother, thereby enabling the student to learn from softer labels and generalizes better.

With few exceptions, our method achieves the best performance. Furthermore, with some exceptions, our method outperforms the traditional FitNet method. From these results, we confirm that our method generally yields better performance than the conventional regressor-based approach. In addition, for ShuffleNetV2, both logit and ILD often exhibit performance drops after distillation, sometimes with large gap, indicating that it is difficult to effectively transfer teacher's information well in this model family. However, our method improves performance in all but two cases. Even in those two cases, the performance drop under our method is less severe than under the other two methods. Taken together, we confirm that our method is a powerful distillation approach.

In particular, distillation at the last layer immediately preceding the classification layer yields dramatic performance improvements in all cases. In most cases, it outperformed all other pruning methods and layer settings by a considerable margin. Remarkably, in ResNet with CIFAR-100 and TinyImageNet, it achieved 77.50%, 49.23%, respectively, exceeding even the teacher's performance.

### 4.3 ABLATION STUDIES

#### 4.3.1 PROBING ON INTERMEDIATE LAYER

As introduced in the preliminary experiments, we used the probing method (Alain & Bengio, 2016) to measure how much information a feature map in an intermediate layer contains. Figure 3 compares the probing results of the teacher, the original student, and students distilled using the traditional ILD method and our method. All results are measured on the feature map of layer 3, which was also selected as the target layer.

Figure 3a shows the results obtained with the standard probing setup, which employs a single-layer FC layer as a probing model. As expected, the teacher achieved the highest accuracy and our method ranked second. This indicates that, when we distill using our method, the target layer in the student retains more information than in either the original student or the student distilled using the traditional ILD method, which is consistent with our main experimental results. In contrast, the student distilled using the traditional ILD method performed even worse than the undistilled original student, which is a phenomenon that we observed not only at this layer but also at several other layers. Because this result contradicted our expectations, we conducted the additional experiment shown in Figure 3b.

For Figure 3b, instead of using the usual shallow single-layer probing model, we adopted a deeper five-layer model. While the shallow probing model mainly evaluates information that is explicitly represented in the feature map, the deeper model was intended to extract information that is more implicitly embedded within the representation (Hewitt & Liang, 2019). The results resemble those in Figure 3a in that the teacher still performs best, and ours comes next. However, unlike in Figure 3b, the student distilled using the traditional ILD method now slightly surpasses the original student. This suggests that the traditional method, which transfers information indirectly through a regressor, primarily conveys implicit information, whereas our method, which distills directly into the student, can pass along more explicit information. Moreover, observing that in both settings our method outperforms the conventional approach, we can conclude that our method transfers a greater amount

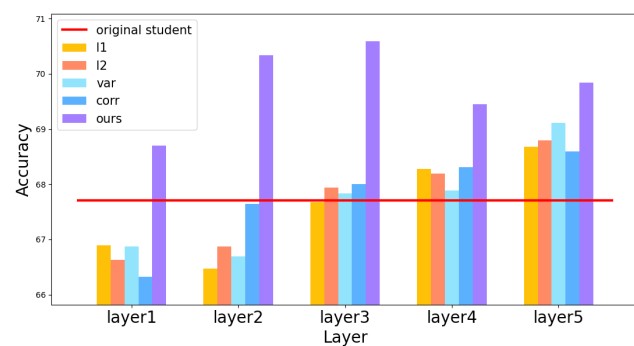

Figure 4: Comparing our method with direct teacher feature map distillation.

of information. We observed a similar pattern emerge in many, though not all, other models and layers.

### 4.3.2 DIRECT FEATURE MAP DISTILLATION WITHOUT TEACHER PRUNING

The teacher is ultimately serves only as a means of distillation, and retraining the teacher after pruning incurs additional overhead. To bypass the teacher retraining stage, we extracted the teacher's feature map, reduced its dimension using several criteria, and directly distilled this feature map into the student. Figure 4 presents the experiments in which the distillation was performed via direct feature map distillation without pruning and retraining. The experiments compare the performance of students distilled from feature maps whose dimensions were reduced by various metrics at each layer in VGG family, against the performance of our method.

The results show that in most cases, our method achieves a higher performance than the other methods. This indicates that the information recovered during retraining plays an important role in distillation and so, teacher pruning is an important step in our method. However, this direct distillation approach has the distinct advantage that it does not require teacher pruning, and thus it can be used even when teacher pruning is unavailable.

## 5 CONCLUSIONS

In our work, we demonstrated, through probing experiments, that the regressor commonly used to resolve the dimension mismatch in traditional intermediate layer distillation is inefficient. To address the dimension mismatch problem without a regressor, we proposed a new distillation method that prunes the teacher to resolve this mismatch and demonstrated, through extensive experiments, that it yields significantly higher performance than traditional approaches that rely on a regressor. Our method also has the advantage that it can be readily combined with existing intermediate layer distillation techniques whenever teacher pruning is available. A limitation of our study is that we evaluated only CNN models and, therefore, did not measure performance on other architectures such as Vision Transformers. In particular, for ViT families, the structure makes it difficult to prune only specific layers of the teacher, so exploring other model families like ViT remains as a subject of future work. In addition, further exploration of the integrated use of knowledge distillation and pruning is left for future work.

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

Table 2: Training hyperparameters. Step($s$, $\gamma$) denotes StepLR scheduler with a step size $s$ and a decay factor $\gamma$.

| Model | CIFAR-100 | | | TinyImageNet | | |
|---|---|---|---|---|---|---|
| | Epochs | LR | Scheduler | Epochs | LR | Scheduler |
| ResNet101 | 200 | 0.10 | Step(50, 0.2) | 200 | 0.10 | Step(50, 0.1) |
| ResNet101 (pruned) | 100 | 0.001 | Step(20, 0.5) | 100 | 0.001 | Step(20, 0.5) |
| ResNet18 | 200 | 0.10 | Step(50, 0.2) | 200 | 0.10 | Step(50, 0.2) |
| VGG16×4 | 200 | 0.10 | Step(50, 0.2) | 200 | 0.10 | Step(50, 0.2) |
| VGG16×4 (pruned) | 100 | 0.10 | Step(20, 0.2) | 100 | 0.10 | Step(20, 0.2) |
| VGG11 | 200 | 0.10 | Step(50, 0.2) | 200 | 0.10 | Step(50, 0.2) |
| ShuffleNetV2×2.0 | 300 | 0.05 | Cosine | 300 | 0.01 | Cosine |
| ShuffleNetV2×2.0 (pruned) | 100 | 0.001 | Cosine | 100 | 0.001 | Cosine |
| ShuffleNetV2×0.5 | 300 | 0.05 | Cosine | 300 | 0.10 | Cosine |

Ruiqing Wang, Shengmin Wan, Wu Zhang, Chenlu Zhang, Yu Li, Shaoxiang Xu, Lifu Zhang, Xiu Jin, Zhaohui Jiang, and Yuan Rao. Progressive multi-level distillation learning for pruning network. *Complex & Intelligent Systems*, 9(5):5779–5791, 2023.

Ying Zhang, Tao Xiang, Timothy M Hospedales, and Huchuan Lu. Deep mutual learning. In *Proceedings of the IEEE conference on computer vision and pattern recognition*, pp. 4320–4328, 2018.

Z Zhang, G Ning, and Z He. Knowledge projection for effective design of thinner and faster deep neural networks. *ArXiv, vol. abs/1710.09505*, 2017.

Borui Zhao, Quan Cui, Renjie Song, Yiyu Qiu, and Jiajun Liang. Decoupled knowledge distillation. In *Proceedings of the IEEE/CVF Conference on computer vision and pattern recognition*, pp. 11953–11962, 2022.

# A  APPENDIX

## A.1  IMPLEMENTATION DETAILS

The hyperparameters used for training are described in Table 2. For data augmentation, we applied only random cropping, random horizontal flipping, and random rotation. All models were trained with a batch size of 128 and optimized using stochastic gradient descent with weight decay of 0.0005 and momentum of 0.9. For logit distillation, we set the temperature to 10, while conventional ILD with a regressor employed the standard MSE loss. In contrast, our method used CosSim loss as noted above. We also evaluated our method with both MSE loss and CosSim loss, with CosSim yielding better performance. Further experimental details on the MSE and CosSim losses are provided in Appendix A.3.2.

## A.2  HARDWARE/SOFTWARE

All experiments were conducted on an Ubuntu 20.04 server with Python 3.8, PyTorch 2.4 and CUDA 12.2, using a single NVIDIA RTX 3090 GPU per run.

## A.3  ADDITIONAL ABLATION STUDIES

### A.3.1  ABLATION STUDIES ON ALPHA IN LOSS FUNCTION

We measured how the performance of our model changes with different values of $\alpha$. Figure 5 shows the performance for each layer of VGG11 distilled by our method from the pruned VGG16x4 under varying values of $\alpha$. All other experimental settings are identical to those of the main experiment and only $\alpha$ was varied. As the results indicate, a smaller $\alpha$ value generally leads to higher final accuracy, which is the case where more weights are placed on the distillation loss than on the classification

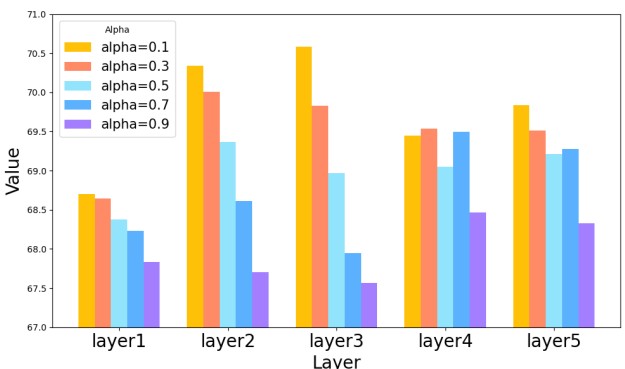

Figure 5: Ablation studies on various $\alpha$ values.

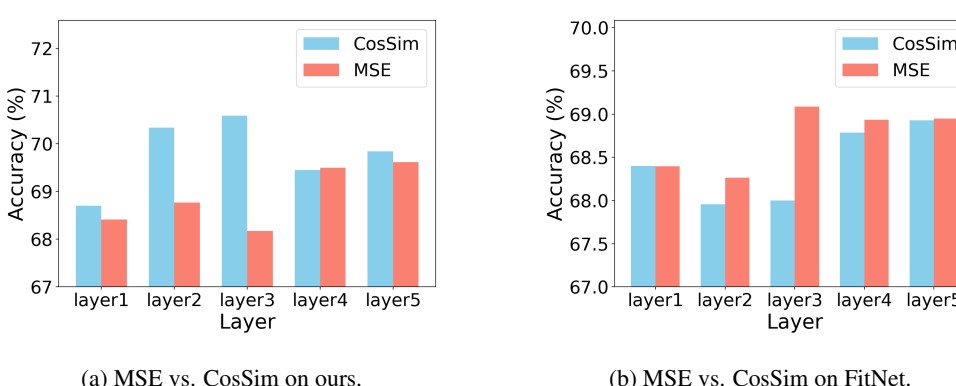

(a) MSE vs. CosSim on ours.

(b) MSE vs. CosSim on FitNet.

Figure 6: Ablation study on the choice of the loss function.

loss. This confirms that the information transferred from the intermediate layer of the teacher has a substantial impact on the classification result, sometimes even exceeding the effect of learning the classification task directly. However, although not shown in the figure, we observed that if $\alpha$ is further reduced, the performance gradually decreases, and when $\alpha$ reaches 0, the classification results become nearly random. This is not surprising, because when $\alpha = 0$ only the target intermediate layer and above are trained, leaving all layers below completely untrained.

### A.3.2 ABLATIONS ON TYPE OF LOSS FUNCTIONS

As mentioned in the previous sections, we used the cosine similarity loss (CosSim) instead of the MSE loss that is commonly employed in ILD. Empirically, the CosSim loss yielded a higher average performance than the MSE loss in our experiments. Figure 6 presents the performance differences between using MSE and CosSim losses.

Figures 6a and 6b compare performance when each layer of the VGG model is chosen as the target layer and training is carried out using MSE or CosSim as the loss function, with Figure 6a reporting results for our method and Figure 6b for FitNet. In our method, every block using CosSim loss achieves a higher performance than when using MSE loss. In contrast, when the same experiments were conducted with the FitNet method, the performance difference was marginal, with MSE yielding slightly higher accuracy. A similar phenomenon is observed in other models and datasets, where the last layer shows comparable performance regardless of the loss used. In other models and datasets, for the layers other than the last one, CosSim loss almost always delivers higher performance, consistent with the above results.

### A.4 LLM USAGE

We disclose that LLM was used in this study exclusively for partial debugging of visualization scripts and grammar checking during paper writing. All generated content was reviewed, rewritten, and validated by the authors.

