# OpenReview forum: "Regressor-free Intermediate Layer Distillation via Teacher Pruning"
_ICLR.cc/2026/Conference — Submitted to ICLR 2026_

### Official Review · Reviewer_4fui · 2025-10-16

**Soundness:** 2
**Presentation:** 2
**Contribution:** 1
**Rating:** 2
**Confidence:** 5

**Summary:**

The paper claims to be the first to apply a pruning-based approach to knowledge distillation that removes the use of a regressor to overcome the parameter mismatch between the teacher and student networks. To support this, the paper claims that they theoretically demonstrate the limitations of using a regressor. Furthermore, the paper claims that the proposed method outperforms existing limited approaches through experiments conducted on small yet basic datasets such as CIFAR-100 and TinyImageNet.

**Strengths:**

- The paper adopts a pruning method to eliminate the need for a regressor in knowledge distillation between homogeneous networks with different dimensions or layers.

- The authors use mutual information to convincingly address the limitations of conventional approaches.

- The paper is simple to read and get the contribution of the paper.

**Weaknesses:**

- The proposed regressor-free method using pruning only when the number of layers is the same and only the number of channels differs. It does not provide a solution for cases where the layer counts differ, and it results in limiting its applicability. Furthermore, it cannot address knowledge distillation between heterogeneous networks.

- Although knowledge distillation is a relatively young field, there already exist numerous studies. The related work section in this paper is insufficiently comprehensive, and the contributions are not clearly distinguished from similar prior works, such as:\
[1] O’Neill et al., Deep Neural Compression via Concurrent Pruning and Self-Distillation, arXiv:2109.15014, 2021\
[2] Aghli et al., Combining Weight Pruning and Knowledge Distillation for CNN Compression, CVPR Workshop 2021\
[3] Muralidharan et al., Compact Language Models via Pruning and Knowledge Distillation, arXiv:2407.14679, 2024

- The performance comparison with previous methods is quite poor. Comparing only with LD, FitNet, and conventional KD does not ensure fairness. The paper should provide the SOTA comparison tables because the main claim of this paper is the traditional regressor-based KD has the limitation and the proposed regressor-free method does not.

- Although the paper claims to demonstrate its superiority through extensive experiments, the evaluation is limited to small-scale toy datasets such as CIFAR-100 and TinyImageNet. These datasets are too small, and CIFAR-100, in particular, is known to produce high variation across runs, making it difficult to ensure objective results. Therefore, more comprehensive experiments using larger datasets such as ImageNet are generally required to validate the proposed method.

**Questions:**

Now I have no question on this paper.

---

> ### Author Response · Authors · 2025-11-21
> **Author Response to Reviewer 4fui**
>
> We sincerely appreciate your careful and constructive review. Your comments have been very helpful in improving the clarity and rigor of our work. We have carefully considered all comments and made corresponding revisions and additions. Below we respond to each point in turn.
>
> 1. The proposed regressor-free method using pruning only when the number of layers is the same and only the number of channels differs. It does not provide a solution for cases where the layer counts differ, and it results in limiting its applicability. Furthermore, it cannot address knowledge distillation between heterogeneous networks
>
>     - In feature-based ILD, the common practice is to align corresponding intermediate layers of teacher and student, applying a regressor when shapes differ as Gou et al[1]. Accordingly, we followed this standard setting. That said, our method is applicable to cross-layer and heterogeneous cases as well, and reflecting the reviewer’s suggestion, we are planning additional experiments along this line. We appreciate the helpful suggestion.
>
>     [1] Gou, Jianping, et al. "Knowledge distillation: A survey." *International journal of computer vision* 129.6 (2021): 1789-1819.
>
> 2. Although knowledge distillation is a relatively young field, there already exist numerous studies. The related work section in this paper is insufficiently comprehensive, and the contributions are not clearly distinguished from similar prior works, such as:
>
>     - In the Related Work section, we introduced studies that address distillation and pruning jointly. The cited papers fall into the category described in the main text as “the model before pruning serves as a teacher and the model after pruning serves as a student, and distillation is performed to preserve performance after pruning.” We will add explicit citations to these works.
>
> 3. The performance comparison with previous methods is quite poor. Comparing only with LD, FitNet, and conventional KD does not ensure fairness. The paper should provide the SOTA comparison tables because the main claim of this paper is the traditional regressor-based KD has the limitation and the proposed regressor-free method does not.
>
>     Although the paper claims to demonstrate its superiority through extensive experiments, the evaluation is limited to small-scale toy datasets such as CIFAR-100 and TinyImageNet. These datasets are too small, and CIFAR-100, in particular, is known to produce high variation across runs, making it difficult to ensure objective results. Therefore, more comprehensive experiments using larger datasets such as ImageNet are generally required to validate the proposed method.
>
>
>     - We are running experiments on ImageNet-100 and will add the results to the paper. We are also conducting comparisons against additional SOTA ILD baselines (AT, CRD, RKD). These experiments are well underway, and we can include the resulting numbers in the camera-ready version.

---

> > ### Comment · Reviewer_4fui · 2025-11-21
> >
> > Fundamentally, rather than providing precise and in-depth theoretical study to justify the novelty of the proposed method, the paper relies on general mathematical expressions to support its novelty. It still fails to dispel my major concerns regarding whether the method is truly novelty in the knowledge distillation fields. A clear and rigorous theoretical study of why the proposed method leads to improvements is necessary, rather than relying solely on empirical study.
> >
> > In addition, as several reviewers have already pointed out, the absence of experiments on large-scale datasets and comparisons with the numerous existing state-of-the-art methods further amplifies these concerns. It is also important that the comparisons with prior methods follow widely used and established evaluation protocols, rather than relying on own protocol with the limited prior methods (not latest ones), in order to ensure fairness and validate the superiority of the proposed method.
> >
> > Based on the current rebuttal provided thus far, there is no compelling justification to change the original rating, and therefore I intend to maintain it.

---

### Official Review · Reviewer_1MZ9 · 2025-10-30

**Soundness:** 2
**Presentation:** 3
**Contribution:** 2
**Rating:** 2
**Confidence:** 5

**Summary:**

This paper proposes a regressor-free intermediate layer distillation method that uses teacher pruning to address dimensional mismatches between teacher and student models. The authors argue that traditional regressor-based approaches lead to indirect and suboptimal knowledge transfer. Through probing experiments, they demonstrate the limitations of regressors and propose pruning the teacher's intermediate layers to align dimensions directly. Extensive experiments on ResNet, VGG, and ShuffleNetV2 across CIFAR-100 and TinyImageNet show that their method often outperforms conventional intermediate layer distillation, sometimes even exceeding the teacher's performance.

**Strengths:**

- The method is simple and applicable across multiple architectures.
- Experimental results show consistent improvements over regressor-based ILD.
- The paper includes ablation studies on loss functions and alpha values.

**Weaknesses:**

**Novelty and Positioning**: The core idea of pruning the teacher for distillation has been explored in Park & No (ECCV 2022). The authors do not clearly distinguish their method beyond the use of intermediate (rather than logit) distillation.

**Theoretical Justification**: The mutual information argument in Section 3.3 is superficial and does not convincingly explain why pruning helps or why mutual information is the right measure for knowledge transfer.

**Experimental Gaps**:
1. No analysis of why VGG on TinyImageNet suffers a significant performance drop after pruning.
2. No explanation for why ShuffleNetV2 is "difficult" for distillation.
3. The retraining cost of the pruned teacher is not accounted for in comparisons.

**Overstated Claims**: The authors claim "comprehensive theoretical analysis" but provide only a brief and non-rigorous inequality. They also claim to "consistently outperform" but simultaneously state they are not targeting SOTA, which is contradictory.

**Questions:**

**Theoretical and Conceptual Concerns**
1. The mutual information-based justification is superficial and lacks rigor in connecting to the actual knowledge transfer process.
2. Mutual information is not adequately justified as a meaningful measure for distillation quality.
3. Theoretical analysis is generally weak and fails to provide a solid foundation for the proposed method.

**Novelty and Positioning**
1. The distinction from prior work, especially Park & No (ECCV 2022), is unclear. Both use teacher pruning before distillation.
2. It is ambiguous whether the core contribution lies only in applying pruning to intermediate layers rather than logits.
3. The related work section does not sufficiently differentiate the method from existing pruning-for-distillation approaches.

**Experimental Analysis and Unexplained Observations**
1. Key experimental phenomena are left unanalyzed: (1) Significant performance drop in VGG on TinyImageNet after pruning. (2) Difficulty in distilling ShuffleNetV2 effectively.
2. The computational cost of pruning and retraining the teacher is not evaluated or compared to baselines (e.g., training time, FLOPs).
3. Experimental gains, while consistent, are not always substantial or thoroughly interpreted.

**Clarity and Consistency of Claims**
1. Some claims are overstated or inconsistent, such as: (1) Stating the goal is not to surpass SOTA, while claiming to “consistently outperform” conventional methods. (2) Describing the theoretical analysis as “comprehensive” when it is minimal.
2. The scope of contributions is not clearly defined relative to the experimental results.

**Methodological and Evaluation Gaps**
1. No analysis is provided on why certain architectures (e.g., VGG, ShuffleNetV2) behave differently under the proposed method.
2. Broader applicability is not tested on modern architectures such as Vision Transformers (ViTs).
3. The impact of architectural elements like skip connections or channel shuffling in ShuffleNetV2 is not discussed.

---

> ### Author Response · Authors · 2025-11-21
> **Author Response to Reviewer 1MZ9**
>
> We sincerely appreciate your careful and constructive review. Your comments have been very helpful in improving the clarity and rigor of our work. We have carefully considered all comments and made corresponding revisions and additions. Below we respond to each point in turn.
>
> 1. **Theoretical and Conceptual Concerns**
>
>     - Section 3.3 establishes, from a mutual-information perspective, a lower bound indicating that our method transfers at least as much knowledge as conventional regressor-based distillation methods.
>
> 2. **Novelty and Positioning**
>
>     - The key distinction of our work from prior papers is that we explicitly identify the limitations of using regressor in ILD and adopt teacher-side pruning as a principled mechanism to remove it.
>
>     - “Prune-then-distill” shows that performing logit distillation after pruning can improve performance. In contrast, we showed that the regressor hinders the performance of ILD and suggested the pruning method to remove the regressor in ILD. To show that pruning is not the sole cause of the performance gain, we conducted a direct head-to-head comparison with “prune-then-distill” (see Table 1), which indicates that the advantage is attributable to eliminating the regressor.
>
> 3. **Experimental Analysis and Unexplained Observations**
>
>     - We acknowledge that our analysis is insufficient regarding the performance drop of VGG on TinyImageNet after pruning and the degradation observed on ShuffleNet after distillation. We will conduct supplementary experiments to provide adequate justification, and we appreciate the reviewer for highlighting these points.
>
>     - Regarding computational cost, pruning itself adds negligible overhead; as reported in Appendix Table 2, the only extra cost comes from a brief retaining phase (a small number of additional epochs) for the pruned teacher. In our current setup this corresponds to roughly a 1.5× increase in *teacher* training cost, with identical inference cost. We will state this more explicitly in the main text.
>
> 4. **Clarity and Consistency of Claims**
>
>     - Our claim is that the proposed method consistently outperforms regressor-based approaches and we did not position this as a comparison against SOTA. That said, we agree that additional comparisons with representative distillation methods are useful, and we will include them.
>
> 5. **Methodological and Evaluation Gaps**
>
>     - As noted above, we will conduct supplementary experiments and analyses to address the observed drops on VGG and ShuffleNetV2.
>
>     - Regarding ViTs, as stated in the conclusion, we leave this to future work; furthermore, an in-depth analysis specific to ShuffleNet’s unique properties is beyond the scope of the current paper.

---

> > ### Comment · Reviewer_1MZ9 · 2025-11-26
> >
> > Thank you for your detailed response. You have adequately addressed some of my concerns.
> > However, several key issues remain unresolved. The theoretical justification, while noted, still lacks the desired rigor. More critically, the unexplained performance drops (VGG on TinyImageNet, ShuffleNet post-distillation) are fundamental to validating the robustness of your method, and their analysis remains pending.
> >
> > While I acknowledge the planned revisions, the core methodological validation is incomplete at this stage. Therefore, I **maintain my original score unchanged**.

---

### Official Review · Reviewer_oZFR · 2025-10-30

**Soundness:** 2
**Presentation:** 2
**Contribution:** 2
**Rating:** 2
**Confidence:** 5

**Summary:**

The authors proposed a regressor-free intermediate layer distillation method. They prune the teacher's target layer (L1 channels) to match the student's, with a retaining process. The experiments show that it is effective on small CNN benchmarks. However, the core idea overlaps with “prune-then-distill” [1] and pruning teacher width to enable ILD as in InDistill [2]. Thus, the novelty is incremental.

[1] Park, J., No, A. “Prune Your Model Before Distill It.” ECCV 2022 / arXiv:2109.14960.
[2] Sarridis, I. et al. “InDistill: Information Flow-Preserving Knowledge Distillation for Model Compression.” WACV 2025 / arXiv:2205.10003.

**Strengths:**

1. The core idea of pruning for feature distillation shall work. The process is straightforward and appears to be easily reproducible.
2. The motivation is well-supported by various tests and visualizations.
3. The experiments are consistent with the motivation, and the implementation details are clearly presented.

**Weaknesses:**

1. The core idea overlaps with “prune-then-distill” [1] and pruning teacher width to enable ILD as in InDistill [2].
2. The current pruning scheme lacks thorough development, and conducting an ablation study on this aspect would be beneficial.
3. As a distillation method, the proposed approach requires retraining the pruned teacher model on the same dataset. This results in high computational costs, potentially rendering it impractical in many scenarios. Additionally, this contradicts the authors' assertion that "the performance gap between the pruned teacher and the original teacher remains negligible."
4. The assumption that "γ can be regarded as negligibly small" does not hold. Consequently, Eq. 6 cannot support the conclusion that the performance of the proposed method is lower-bounded by that of traditional regressor-based ILD methods. Meanwhile, should it be I(f_t, f_s) instead in Eq. 5?
5. All experiments are limited to CIFAR-100 and TinyImageNet datasets, with no results provided for ImageNet-1K or CIFAR-10. This limitation hinders the ability to evaluate practical impact and robustness at realistic resolutions effectively. Furthermore, many state-of-the-art methods have not been included in the comparisons.

[1] Park, J., No, A. “Prune Your Model Before Distill It.” ECCV 2022 / arXiv:2109.14960.
[2] Sarridis, I. et al. “InDistill: Information Flow-Preserving Knowledge Distillation for Model Compression.” WACV 2025 / arXiv:2205.10003.

**Questions:**

1. Please discuss the differences between "prune-then-distill" [1] and InDistill [2].
2. The motivation and assumptions require further investigation.

[1] Park, J., No, A. “Prune Your Model Before Distill It.” ECCV 2022 / arXiv:2109.14960.
[2] Sarridis, I. et al. “InDistill: Information Flow-Preserving Knowledge Distillation for Model Compression.” WACV 2025 / arXiv:2205.10003.

---

> ### Author Response · Authors · 2025-11-21
> **Author Response to Reviewer oZFR**
>
> We sincerely appreciate your careful and constructive review. Your comments have been very helpful in improving the clarity and rigor of our work. We have carefully considered all comments and made corresponding revisions and additions. Below we respond to each point in turn.
>
> 1. The core idea overlaps with “prune-then-distill” [1] and pruning teacher width to enable ILD as in InDistill [2].
>
>     - The key distinction of our work from prior papers is that we explicitly identify the limitations of using regressor in ILD and adopt teacher-side pruning as a principled mechanism to remove it.
>
>     - “prune-then-distill” shows that performing logit distillation after pruning can improve performance. In contrast, we showed that the regressor hinders the performance of ILD and suggested the pruning method to remove the regressor in ILD. To show that pruning is not the sole cause of the performance gain, we conducted a direct head-to-head comparison with “prune-then-distill” (see Table 1), which indicates that the advantage is attributable to eliminating the regressor.
>
>     - As noted in our paper (pruning was introduced only as an auxiliary step within the overall pipeline, rather than as a primary methodology for ILD, and the design or independent effectiveness of teacher pruning for ILD was not analyzed in depth.), InDistill does not address the regressor issue. While the low-level implementation may appear similar, the motivation for using pruning and the resulting conclusions are fundamentally different.
>
> 2. The current pruning scheme lacks thorough development, and conducting an ablation study on this aspect would be beneficial.
>
>     - Our objective is to realize ILD without a regressor. Therefore, the specific choice of teacher-side pruning criterion is not central to the contribution, and any reasonable pruning method should suffice. For this reason, we deliberately adopted the most straightforward option, which is L2-norm for channel pruning. That said, we agree that examining sensitivity to the pruning rule is informative. We appreciate the suggestion and will incorporate an ablation comparing alternative criteria in the revision.
>
> 3. As a distillation method, the proposed approach requires retraining the pruned teacher model on the same dataset. This results in high computational costs, potentially rendering it impractical in many scenarios. Additionally, this contradicts the authors' assertion that "the performance gap between the pruned teacher and the original teacher remains negligible."
>
>     - Regarding computational cost, pruning itself adds negligible overhead; as reported in Appendix Table 2, the only extra cost comes from a brief retaining phase (a small number of additional epochs) for the pruned teacher. In our current setup this corresponds to roughly a 1.5× increase in *teacher* training cost, with identical inference cost. We will state this more explicitly in the main text.
>
> 4. The assumption that "γ can be regarded as negligibly small" does not hold. Consequently, Eq. 6 cannot support the conclusion that the performance of the proposed method is lower-bounded by that of traditional regressor-based ILD methods. Meanwhile, should it be I(f_t, f_s) instead in Eq. 5?
>
>     - Our statement that “γ can be regarded as negligibly small” is empirically supported (Our experiments show almost no performance gap between the original and pruned teachers, so γ can be regarded as negligibly small.).
>
>     - The issue in Equation 5 is indeed our mistake; we appreciate the reviewer for pointing it out and will correct it accordingly.
>
> 5. All experiments are limited to CIFAR-100 and TinyImageNet datasets, with no results provided for ImageNet-1K or CIFAR-10. This limitation hinders the ability to evaluate practical impact and robustness at realistic resolutions effectively. Furthermore, many state-of-the-art methods have not been included in the comparisons.
>
>     - We are running experiments on ImageNet-100 and will add the results to the paper. We are also conducting comparisons against additional SOTA ILD baselines (AT, CRD, RKD). These experiments are well underway, and we can include the resulting numbers in the camera-ready version.

---

### Meta-Review · Area_Chair_np8o · 2026-01-07

**Summary:**

It received three "2: reject" ratings so cannot be accepted.

**Reviewer Concerns:**

The main concerns are novelty and sufficient experiments. For instance, the experiments are limited to CIFAR100 and TinyImagenet only. When a reviewers asks for larger datasets, the authors said they will add ImageNet100 later which may not be still large enought to empirically prove that the idea works.

**Reviewer Scores:**

It got three "2" ratings. Two of the reviews got the chance to respond to the rebuttal and were not convinced by the rebuttal. One of the reviewers could not respond, but I do not think they would have changed their rating since the authors are suggesting to add experiments on ImageNet100, which in my view may not be still large enough to empirically prove the effectiveness of the proposed method.

---

### Decision · Program_Chairs · 2026-01-26

Reject